# Complete De Novo Assembly of *Wolbachia* Endosymbiont of *Frankliniella intonsa*

**DOI:** 10.3390/ijms241713245

**Published:** 2023-08-26

**Authors:** Zhijun Zhang, Jiahui Zhang, Qizhang Chen, Jianyun He, Xiaowei Li, Yunsheng Wang, Yaobin Lu

**Affiliations:** 1State Key Laboratory for Managing Biotic and Chemical Threats to the Quality and Safety of Agro-Products, Institute of Plant Protection and Microbiology, Zhejiang Academy of Agricultural Sciences, Hangzhou 310021, China; 15096100996zjh@gmail.com (J.Z.); 17300906030@163.com (Q.C.); hjy12242021@163.com (J.H.); lixiaowei1005@163.com (X.L.); lvyb@zaas.ac.cn (Y.L.); 2Hunan Provincial Key Laboratory for Biology and Control of Plant Diseases and Insect Pests, College of Plant Protection, Hunan Agricultural University, Changsha 410128, China

**Keywords:** *Wolbachia*, *Frankliniella intonsa*, symbiosis, phylogeny, comparative genomics

## Abstract

As an endosymbiont, *Wolbachia* exerts significant effects on the host, including on reproduction, immunity, and metabolism. However, the study of *Wolbachia* in Thysanopteran insects, such as flower thrips *Frankliniella intonsa*, remains limited. Here, we assembled a gap-free looped genome assembly of *Wolbachia* strain *w*FI in a length of 1,463,884 bp (GC content 33.80%), using Nanopore long reads and Illumina short reads. The annotation of *w*FI identified a total of 1838 protein-coding genes (including 85 pseudogenes), 3 ribosomal RNAs (rRNAs), 35 transfer RNAs (tRNAs), and 1 transfer-messenger RNA (tmRNA). Beyond this basic description, we identified mobile genetic elements, such as prophage and insertion sequences (ISs), which make up 17% of the entire *w*FI genome, as well as genes involved in riboflavin and biotin synthesis and metabolism. This research lays the foundation for understanding the nutritional mutualism between *Wolbachia* and flower thrips. It also serves as a valuable resource for future studies delving into the intricate interactions between *Wolbachia* and its host.

## 1. Introduction

*Wolbachia*, gram-negative bacteria, were first discovered in the reproductive tissue of *Culex pipiens* in 1924 and officially named *Wolbachia pipientis* in 1936 [1,2]. These matrilineally inherited endosymbionts are widely distributed among arthropods and nematodes [3]. *Wolbachia* are primarily transmitted in their host by vertical and horizontal modes [4]. Vertical transmission, via cytoplasmic transfer from the mother’s germ cell to the offspring, is the primary mode of transmission, while horizontal transmission occurs between different hosts.

The wide distribution of *Wolbachia* can be attributed to its ability to be efficiently transmitted maternally and to manipulate host reproduction in a way that benefits infected females [5]. *Wolbachia* play a critical role in regulating the reproductive phenotype of their host through various mechanisms, including cytoplasmic incompatibility (CI), parthenogenesis induction (PI), feminization (FEM), and male killing (MK). In arthropods, CI is the most common phenotype induced by *Wolbachia*, resulting in embryonic death when uninfected females mate with infected males or males and females carrying incompatible *Wolbachia*. The *cifA* and *cifB* genes associated with phage WO are of significant importance in the study of CI, with their homologs classified as Types I-V [6]. In addition, *Wolbachia* can induce parthenogenesis in haplodiploid insects, leading to the production of female rather than male offspring from unfertilized eggs. Furthermore, *Wolbachia* influences the morphological expression of female characteristics in male offspring and can selectively eliminate developing *Wolbachia*-infected males thereby consequently altering the male–female ratio.

In *Drosophila melanogaster*, the presence of *Wolbachia* has been shown to confer resistance to RNA viruses such as *Drosophila* C virus, Nora virus, and Flock House virus, the first two of which are natural pathogens of *D. melanogaster* [7]. Conversely, when *w*MelPop from *D. melanogaster* was introduced into *Aedes aegypti* mosquitoes, it resulted in a reduction in the lifespan of adult female mosquitoes [8]. In addition, *Wolbachia* has the ability to infect thrips and regulate their reproductive phenotype. For example, it mediates parthenogenesis in *Franklinothrips vespiformis* and induces CI in *Pezothrips kellyanus* [9,10]. Comparative genomic studies have identified several genes associated with CI modification and rescue functions [11]. Remarkable transcriptional differences have been observed in the *cifA* and *cifB* genes during host development, and these genes may be degraded or lost in *Wolbachia* strains that no longer induce CI [12]. While most *Wolbachia* infections in arthropods are considered facultative mutualists, meaning that the host can survive and reproduce without the presence of *Wolbachia*, there are cases where *Wolbachia* act as obligate mutualists. For example, bed bugs rely on *Wolbachia* to provide essential B vitamins for their normal development and reproduction. Removing *Wolbachia* from bed bugs with antibiotics can lead to impaired growth and sterility [13]. Additionally, genome analysis of *Wolbachia* in planthoppers has revealed the presence of intact biotin and riboflavin biosynthetic operons. This suggests that *Wolbachia* have the ability to synthesize biotin and riboflavin, potentially enhancing the reproductive capacity of the host [14].

Initially, the classification of *Wolbachia* was primarily based on the 16S rRNA and *wsp* genes [15,16], which encode *Wolbachia* surface proteins. However, 16S rRNA, with its high conservation and slow evolutionary rate, cannot effectively distinguish closely related strains. In addition, the high recombination and strong diversifying selection of the *wsp* gene has led to potential confusion in strain clustering and introduced bias. To address these issues, MLST was introduced as an effective tool for *Wolbachia* classification [17]. The combination of alleles from five conserved genes (*ftsZ*, *gatB*, *coxA*, *hcpA*, and *fbpA*) in *Wolbachia* was used as the MLST to assign a sequence type (ST) to the *Wolbachia* strain. Each ST represented a unique genetic allelic profile characterizing a particular strain of *Wolbachia*. The relevant information on the five conserved genes and their locus, as well as details of the STs, can be accessed via the PubMLST database [18].

*Wolbachia* strains from different hosts typically belong to different supergroups. *Wolbachia* of supergroups A and B are known to infect arthropods exclusively, with supergroup A being predominantly found in Diptera and Hymenoptera, whereas supergroup B is mainly found in Lepidoptera [19]. Conversely, supergroups C, D, and J are exclusively associated with *Wolbachia* infecting filarial nematodes, as supported by several studies [20,21,22]. In addition, supergroup F *Wolbachia* have been found to infect both arthropods and nematodes [23]. Meanwhile, supergroup L *Wolbachia* have been reported to exclusively infect plant-parasitic nematodes [24,25]. Notably, supergroups G and R are no longer considered to be separate entities, with the former considered to be recombinants of supergroup A and B, while the latter is considered to be part of supergroup A [26,27]. Other supergroups of *Wolbachia* are predominantly found in arthropods [28,29,30,31,32]. Supergroup A and B *Wolbachia* have been observed to infect and coexist within the same arthropod hosts, highlighting the irreversible separation of supergroups. Similarly, supergroup B and K have also been reported to coexist within the same host [31,33].

*Wolbachia* has previously been detected in flower thrips, *F. intonsa*, and the housekeeping genes (*gatB*, *coxA*, *hcpA*, *ftsZ*, and *fbpA*) have been cloned for MLST, resulting in ST code 397 [34]. However, despite this previous work, there has been limited investigation of the relevant genomic information of *Wolbachia* in *F. intonsa* (*w*FI). This study presents the first complete genome of *Wolbachia* in *F. intonsa*, which will serve as a valuable resource for future investigations into the functional role of this endosymbiont in flower thrips.

## 2. Results

### 2.1. Assembly and Annotation

We have obtained the genome assembly of the *Wolbachia* strains that infect *F. intonsa*. The genome of *Wolbachia* strain *w*FI was 1,463,884 bp in size and had a GC content of 33.80%. A total of 1877 genes were identified, including 1838 protein-coding genes with 85 candidate pseudogenes, 3 rRNA genes, 35 tRNA genes, and 1 tmRNA (Table 1). The BUSCO completeness scores of the *w*FI genome indicated that it contained 183 complete and single-copy BUSCO groups, 3 complete and duplicated BUSCO groups, 2 fragmented BUSCO groups, and 31 missing BUSCO groups. The BUSCO completeness of the *w*FI genome score of 85.0% was comparable to that of the other two *Wolbachia* complete genomes belonging to supergroup B (Appendix A). The genomic features of *w*FI, including coding sequences (CDS), tRNA, rRNA, and others, were visualized using the CGView Server (Figure 1 and Appendix A). The presence of the irregular GC skew in *w*FI, meant that there was no obvious GC skew to identify a possible origin of replication, which also suggested that *w*FI may have undergone frequent rearrangements. This abnormal GC skew pattern has also been observed in other *Wolbachia* genomes [35,36,37,38,39]. However, it has been observed that *Wolbachia* with reduced genomes exhibit a strong GC skew [40].

A total of 1564 protein-coding genes with Clusters of Orthologous Genes (COG) functional annotations were identified in the *w*FI genome by searching against the eggNOG database v5.0.2 [41]. Of these, 713 (45.58%) genes were classified under the functional category of replication, recombination, and repair (Figure 2A, Appendix A). A total of 486 genes were categorized into 32 different Gene Ontology (GO) terms (Figure 2B). These GO terms were further grouped into three main categories: Biological Process, Cellular Component, and Molecular Function. The Biological Process category consisted of 19 distinct elements with 434 genes. The Cellular Component category included 2 elements, with a total of 401 genes assigned to them. Within the Molecular Function there were 11 elements, with 415 genes assigned to them. Notably, the number of genes associated with cellular process, metabolic process, and the cellular anatomical entity GO term was approximately 400. A total of 540 genes were mapped to 28 different Kyoto Encyclopedia of Genes and Genomes (KEGG) pathways (Figure 2C). Among these pathways, the Metabolism pathway stood out with 309 genes, representing a significantly higher number of coding genes compared to other pathways. Within the metabolism pathway, several subcategories were identified. These included energy metabolism with 96 genes, metabolism of cofactors and vitamins with 75 genes, and carbohydrate metabolism with 64 genes. Furthermore, the genetic information processing pathway included a total of 172 genes, of which 83 genes were associated with translation, 56 genes were involved in replication and repair processes, 34 genes were associated with folding, sorting and degradation, and 3 genes were involved in transcription.

Genome-wide pathway annotation based on the eggNOG database v5.0.2 revealed that the *w*FI genome lacked key genes involved in biotin synthesis, including *bioA*, *bioD*, *bioC*, *bioH*, *bioF*, and *bioB*. However, it contained the *bioY* gene (ko: K03524), which is responsible for biotin transport, and the *birA* gene (ko: K03523), which is involved in the synthesis of essential metabolites using biotin as a precursor. Moreover, we found that, while *w*FI encodes riboflavin synthesis operons, it was not a compact operon but scattered throughout the *Wolbachia* genome, as *ribA*, *ribD*, *ribB*, *ribH*, *ribE*, and *ribF* (Appendix A). The hypothetical phosphatase responsible for coverting 5-amino-6-ribitylamino-2,4(1H,3H)-pyrimidinedione5′-phosphate into 5-amino-6-ribitylamino-2,4(1H,3H)-pyrimidinedione was absent in Appendix A, which may be due to the fact that it remains uncharacterized in most organisms [42].

Pfam annotation of the *w*FI genome revealed 1352 protein-coding genes containing at least one Pfam domain (Appendix A). The most abundant Pfam profile identified was the endonuclease of the DDE superfamily DDE_3 (PF13358). In addition, the domain related to mobile genetic elements, such as DDE_Tnp_1 (PF01609), DDE_Tnp_1_3 (PF13612), DDE_Tnp_1_5 (PF13737), DDE_Tnp_1_IS240 (PF13610), and HTH_Tnp_IS630 (PF01710), were found abundantly in *w*FI genome, which are transposase domains. Moreover, the retroviral integrase domain rve (PF00665), rve_2 (PF13333), rve_3 (PF13683), reverse transcriptase domain RVT_1 (PF00078), and Group II intron reverse transcriptase domain GIIM (PF08388) were identified in the *w*FI genome. Members of the IS3 family have shown remarkable similarity to integrase [43]; this similarity was also shown in *w*FI. The results of the Pfam and IS annotation (Appendix A) revealed that most of the proteins containing retroviral integrase domains were identified as IS3 family members. The Group II intron encoded proteins with reverse transcriptase, maturase, and endonuclease activities. It was observed that introns could mediate genomic rearrangements in the *Wolbachia* genome [44]. This suggested that intron-mediated genomic rearrangement events might be present in *w*FI.

### 2.2. Insertion Sequences, Ankyrin Repeat, Type IV Secretory System, and Prophage Genes

ISs are a critical component of bacterial genomes and consist of inverted repeats and transposase-encoding sequences. They are the simplest mobile genetic elements and play a crucial role in bacterial evolution. To date, approximately 20 families of IS have been identified [45]. *Wolbachia* genomes are known to contain numerous ISs, accounting for approximately 10% of the genome, and play a critical role in *Wolbachia* evolution [46,47]. In this study, 587 open reading frames (ORFs) associated with ISs belonging to 15 IS families were identified in the *w*FI genome. Among them, 80 were considered as putative complete ORFs, 410 as putative partial ORFs, and 97 remained uncategorized ORFs. Notably, the number of IS-associated ORFs in *w*FI was higher compared to other supergroup B *Wolbachia* genomes like *w*Di (GCA_019355355.1) and *w*AlbB (GCA_004171285.1) (Appendix A). The ISs had a median size of 357 bp, ranging from 93 to 1389 bp, and a total of 254.637 bp, accounting for around 17% of the entire *w*FI genome (Appendix A). Among the ISs, the largest family identified was IS 630 with 400 copies (Appendix A).

The Ankyrin repeat (ANK) protein family has been shown to play a critical role in mediating protein-protein interactions and is of particular interest in the context of host-*Wolbachia* interactions. Examination of Pfam protein domains within the *w*FI genome revealed the presence of 38 proteins containing at least one ANK domain, comprising 61 ANK domains with diverse functions (Appendix A).

The Type IV secretion system (T4SS) is a pivotal mechanism used by *Wolbachia* to transfer DNA and/or proteins to eukaryotic cells, which is crucial for successful host infection and proliferation [48,49]. Two operons associated with T4SS have been identified in the *Wolbachia* genome. The first operon comprises *virB8*, *virB9*, *virB10*, *virB11*, *virD4*, and the downstream *wspB* locus, while the second operon includes *virB3*, *virB4*, *virB6*, and several open reading frames (*orf1* to *orf4*) [50]. By combining the prokka, pfam, and eggNOG annotation results and then matching them with NR database, the *w*FI genome revealed the presence of 13 genes associated with T4SSs (Appendix A). These genes were organized into two operons in the *w*FI genome. Furthermore, three duplicated genes, *virB4-2* (*AJACLIMF_00502*), *virB8* (*AJACLIMF_01223*), and *virB9* (*AJACLIMF_01675*), were found scattered elsewhere in the genome.

The majority of the sequenced *Wolbachia* genomes contain the prophage WO, except for those involved in obligate mutualistic relationships [51]. Prophage genes, which are dynamic elements that mediate gene transfer, play a critical role in the evolution and adaptation of the *Wolbachia* genome [52,53]. Notably, these genes are widely distributed throughout the *Wolbachia* genome. The *w*FI genome does not support the notion that each *Wolbachia* genome contains at least one intact prophage WO and usually several degenerate, independently acquired WO prophages [54]. In the *w*FI genome, PHASTER analysis revealed an incomplete prophage region (position: 955,379–965,628) measuring 10.2 kb and consisting of 12 proteins. The incomplete prophage region in *w*FI consisted of two attachment sites (attL and attR), four transposases, a lytic transglycolase, an endonuclease, a transferase, a phage portal protein, one ParE toxin of the type II toxin-antitoxin system, one gpw residue protein, and two uncharacterized proteins. The prophage region of *w*FI, which was incomplete, lacked the modules responsible for encoding head, baseplate, and tail-associated proteins. As a result, it was considered cryptic, meaning that it no longer possessed the ability to form virions and lyse host cells. Additionally, through functional annotation, we identified several prophage-related genes outside of the incomplete prophage region. These genes encode products including phage integrase (*AJACLIMF_00069*, *AJACLIMF_01338*), phage gp6-like head-tail connector protein (*AJACLIMF_01293*), phage tail tube protein (*AJACLIMF_01290*), phage portal protein (*AJACLIMF_01198*, *AJACLIMF_01350*, *AJACLIMF_01351*, *AJACLIMF_01719*), and phage terminase large subunit (*AJACLIMF_01354*) (Appendix A). Prophage was integrated into the *w*FI genome, it could be threatened by selective pressure to lose the gene components required for infection and/or genome replication and scattered on the *w*FI genome as prophage remnants.

### 2.3. Orthology Analysis and Identification of a Core Proteome across Completed Wolbachia Genomes

Using Orthofinder v2.5, we analyzed the orthology relationships between the *w*FI genome and 25 complete *Wolbachia* genomes (Appendix A). Our analysis revealed 1786 orthogroups of 32,733 proteins (Appendix A). Among these orthogroups, 604 were shared by all *Wolbachia* genomes, of which 502 were single-copy orthogroups (Appendix A). Furthermore, orthology analysis identified 79 orthogroups containing 379 proteins that were unique to each individual *Wolbachia* genome analyzed. Additionally, 739 genes were not assigned to any orthogroup (Appendix A). For *w*FI, we assigned 1783 (97.0%) of its protein-coding genes to 973 orthogroups, of which 191 (10.4%) were assigned to 21 species-specific orthogroups, whereas 55 protein-coding genes were not assigned to any orthogroup (Appendix A). The assignment rate for protein-coding genes in other *Wolbachia* genomes was higher than 91.4%, the *w*Cle, which had 150 (11.5%) protein-coding genes that were not assigned to any orthogroup (Appendix A). The core proteome, consisting of 604 common orthogroups that included all proteins present in the analyzed genomes, was significantly higher than the number of orthogroups among other species (Figure 3).

A total of 604 common orthogroups were identified, consisting of 16,224 genes. Among the top 10 enriched GO terms, the ATP metabolic process stood out with 460 genes involved, while regulation of translation and regulation of cellular amide metabolic process had the fewest genes, with 361 each (Figure 4A). As for the top 10 enriched KEGG pathways, Protein export showed the highest number of enriched genes (419), closely followed by Mismatch repair (378) and DNA replication (353) (Figure 4B). In addition, a total of 21 *w*FI-specific orthogroups, containing 191 genes were identified, with the majority of these genes being ISs and belonging to COG L genes. During the enrichment analysis, we found that no genes related to any GO terms were enriched, and none of the genes could be mapped to any specific pathways.

### 2.4. wFI Belong to Supergroup B

The phylogenetic analysis, based on the concatenated sequences of allele of the MLST locus, revealed that *w*FI (ST-397) formed a distinct clade with other ST codes belonging to supergroup B, indicating its classification within this supergroup (Figure 5). To eliminate the interference caused by abnormal *Wolbachia* genomes, we performed a thorough examination of the GC content and genome size. As a result, we identified eight candidate abnormal *Wolbachia* genomes (Appendix A). Further investigation in the NCBI led us to exclude three of the eight candidate abnormal *Wolbachia* genomes. Moreover, two of the eight candidate abnormal *Wolbachia* genomes with an unknown host were also excluded (Appendix A). This resulted in a total of 76 complete genomes, 22 chromosome-scale genomes, 48 scaffold-scale genomes, and 1218 contig-scale genomes (Appendix A). The majority of the 1364 *Wolbachia* genomes belonged to strains infecting the hosts *D. melanogaster* and *D. simulans*, both of which are classified under supergroup A. Using these genomes and *w*FI, we constructed a large-scale NJ phylogenetic tree, which showed that *w*FI clustered with the *Wolbachia* genome belonging to supergroup B (Appendix A). The *Wolbachia* from different supergroups can infect the same host [55]. *Wolbachia* infecting *A. aegypti*, *A. albopictus*, *Dactylopius coccus*, *D. simulans*, and *Nasonia vitripennis* could be categorized into two supergroups, A and B (Appendix A), highlighting the distinct and irreversible segregation of supergroups as evidenced by the co-infection of the same host species by *Wolbachia* belonging to different supergroups. We would like to acknowledge the presence of supergroup classification unknowns, namely *w*Pcr (GCA_918697765.1), *w*Pch (GCA_918342435.1), and *w*Cas (GCA_918231965.1), which infect *Phyllotreta cruciferae*, *Psylliodes chrysocephala*, and *Ceutorhynchus assimilis*, respectively. Notably, *w*Pcr and *w*Pch clustered with supergroup A, whereas *w*Cas clustered with supergroup B, indicating their potential supergroup classification (Appendix A). We used concatenated protein sequences of single-copy genes from 26 complete *Wolbachia* genomes to construct a maximum-likelihood phylogenetic tree using IQ-TREE v2.2.0.3. The resulting tree indicated that the *w*FI formed a clade with the supergroup B *Wolbachia* strains (Figure 6).

In prokaryotes, a commonly used threshold for defining species is a 95% average nucleotide identity (ANI), and an ANI greater than 95% usually indicates that organisms belong to the same species. Conversely, the ANI between different species is typically less than 83% [56]. Notably, the ANI between *w*CfeT and other *Wolbachia* genomes was approximately 78% (Figure 7), indicating that *w*CfeT may have ancestral origins in most other *Wolbachia lineages* [57]. The ANI between *w*FI and *Wolbachia* in supergroup B was approximately 95% (Figure 7), providing compelling evidence that *w*FI belongs to this particular supergroup, which is typically found in arthropods [28].

## 3. Discussion

ISs have been shown to cause large-scale rearrangements in the genome of *Escherichia coli* [58]. In the case of *w*FI, ISs represented a significant proportion, accounting for 17% of the entire genome. Notably, a considerable number of COG L genes, involved in DNA replication, recombination, and repair, were associated with IS elements (Appendix A). The prevalence of ISs in *w*FI suggested that ISs may have caused rearrangements in the *w*FI genome, potentially contributing to the abnormal GG skew observed.

Biotin is a part of the vitamin B family and is not normally synthesized by insects. In many cases, insects rely on microbial symbionts to meet their vitamin B requirements [59]. Studies have shown that *w*Cle (GCA_000829315.1) possesses a complete biotin pathway for vitamin B7, which is essential for the growth and reproduction of its host, the bed bug [60]. Additionally, a complete vitamin B7 synthesis operon was discovered in *w*Nfla (GCA_001675695.1) and *w*Nleu (GCA_001675715.1) [61], while the complete biotin and riboflavin biosynthesis operons were identified in *w*Lug (GCA_007115045.1) and *w*Stri (GCA_001637495.1) [14]. A comprehensive genomic survey investigating the B vitamin synthetic capabilities of various insect-associated *Wolbachia* strains revealed that the riboflavin synthesis pathway was the only one highly conserved pathway [62]. According to the EggNOG annotation results, the biotin synthesis pathway was incomplete in *w*FI. However, despite this deficiency, the *w*FI genome contained the genes *bioY* and *birA*, suggesting that *w*FI could potentially interact with its host or other endosymbionts to provide biotin to the host. The *w*FI possessed a complete pathway of riboflavin, suggesting that *w*FI could have provided riboflavin to the host as a mutualist (Appendix A).

The large-scale NJ tree was able to classify several *Wolbachia* genomes whose supergroup classification was previously unknown. Specifically, *w*Pcr and *w*Pch clustered with supergroup A, whereas *w*Cas clustered with supergroup B (Appendix A). Notably, although WOLB1015 (GCA_902646985.2) and *w*MoviF (GCA_023661065.1) belong to supergroup F and are both presented in the insect *Melophagus ovinus*, they did not cluster together in the same clade in the NJ tree and had an ANI of approximately 99%. In addition, the *w*Chem (GCA_014771645.1) from supergroup T was found to be closely related to *Wolbachia* strains supergroups F, S and D (Appendix A), which is consistent with previous research [63]. The mashtree v1.2.0 [64] allows the rapid clustering of *Wolbachia* genomes into NJ trees, which can provide a first indication of the supergroup classification of unknown *Wolbachia* genomes, and help to identify misclassifications or relationships between supergroups. The classification results obtained using mashtree v1.2.0 [64], *w*FI belong to supergroup B, which is consistent with the results of a previous MLST analysis [34].

*w*FI belong to supergroup B by using mashtree v1.2.0 [64] is consistent of previous MLST analysis result [34].

## 4. Materials and Methods

### 4.1. Genome Assembly

In 2017, adults of *F. intonsa* were collected from *Capsicum annuum* L. plants in Jiaxing, Zhejiang, China (30.75° N, 120.79° E). These insects were subsequently reared on fresh cowpea (*Vigna unguiculata* ssp. *Sesquipedalis*) under controlled conditions at 25 ± 1 °C with a 16 h light cycle. A total of approximately 200 adult *F. intonsa*, representing a mixture of ages, were selected from the laboratory population and subjected to decontamination procedures. Briefly, the adult *F. intonsa* were immersed in 1% sodium hypochlorite solution for 5 min, followed by rinsing with sterile water, then immersion in 70% ethanol, and finally another rinse with sterile water. Subsequently, the samples were rapidly frozen using liquid nitrogen and preserved at −80 °C. The *F. intonsa* genomic DNA was extracted and purified using the QIAGEN DNA tissue kit (QIAGEN 69506, Hilden, Germany), and prepared for sequencing libraries following the manufacturer’s guidelines for sequencing technology (Nextomics Biosciences Co., Ltd., Wuhan, China). Long DNA fragments were sequenced on the Oxford Nanopore PromethION platform, and short-read sequencing was performed on the Illumina NovaSeq 6000 platform.

The *F. intonsa* genome assembly was performed using NextDenovo v2.5.0 [65] with clean Nanopore reads. The assembled sequences were further corrected using NextPolish v1.4.0 [66] with long sequencing reads and Illumina short reads in two iterations, following default parameters. A careful analysis of the *F. intonsa* genome was performed to identify any potential bacterial contamination. During the investigation, a particular looped contig raised suspicion. To gain a clearer understanding, manual binning was carried out using Anvi’o v7 [67], which definitively confirmed that the looped contig corresponded to the *Wolbachia* endosymbiont of *F. intonsa*, specifically identified as *w*FI.

### 4.2. Genome Annotations and Assessments

The identification and annotation of genetic elements are critical for understanding the functional properties of a genome. Here, we employed several bioinformatics tools to identify protein-coding genes as well as non-coding RNA components, including 5S rRNA, 16S rRNA, 23S rRNA, tRNA, and tmRNA. Specifically, we used prokka v1.14.6 [68] to identify these genetic elements with the default parameters. It is important to note that information regarding candidate pseudogenes was mentioned in the log file rather in the result file. To evaluate the completeness of genome, we employed Benchmarking Universal Single-Copy Orthologs (BUSCO) v5.4.3, based on the Proteobacteria_odb10 database [69]. Functional annotation of protein-coding genes was performed by searching against the eggNOG database v5.0.2 [41] using eggNOG-Mapper v2.1.9 [70]. The annotation was performed using the—tax_scope Bacteria parameter to focus on bacterial annotations specifically. Additionally, Pfam domains were annotated using the pfam_scan.pl v1.6 script to search against Pfam database v35.0 [71]. In addition, we employed the PHASTER web server [72] to identify prophage regions within the genome. Furthermore, the ISsaga web server [73] was utilized to identify IS elements presented in the genome. Finally, we used the CGview Server [74] to represent the IS elements, ANK, T4SS, and prophage regions in a circle map.

### 4.3. Comparative Genomics Analysis

The complete assembly of *w*FI was compared to 25 other complete genomes, as listed in Appendix A. All genomes were re-annotated using prokka v1.14.6 [68] with the default parameters. Functional annotation of the protein-coding genes was conducted by searching against the eggNOG database v5.0.2 [41] using eggNOG-Mapper v2.1.9 [70] with the -tax_scope Bacteria parameter. The ANI between *w*FI and 25 complete genomes was calculated using fastANI v1.33 [56]. Orthogroups shared between wFI and 25 complete genomes were identified using Orthofinder v2.5.4 [75], and the common orthogroups across multiple genomes were visualized as UpSet plots [76] using the R package ComplexUpset v1.3.3 [77]. Enrichment analysis for common orthogroups was performed using clusterProfiler V4.6.2 [78].

### 4.4. Phylogenetic Analysis

A total of 12 STs, derived from *Wolbachia* strains belonging to supergroups A, B, D, F, and H, were analyzed (Appendix A). To construct a robust phylogenetic tree, concatenated allelic sequences of the five MLST genes were retrieved from the PubMLST database [18] and aligned using Muscle v5.1 [79]. Poorly aligned regions were then trimmed using Trimal v1.4.1 [80]. The resulting alignment was then subjected to maximum-likelihood analysis using IQ-TREE v2.2.0.3 [81] with ultrafast bootstrap mode and 5000 iterations. The resulting tree was used to classify the *w*FI supergroup. Finally, the maximum-likelihood phylogenetic tree was visualized using ITOL v6 [82].

The NCBI dataset command line tools were used to obtain all *Wolbachia* genome sequences from the GenBank database. Scatter plots based on GC content and genome size were used to identify candidate abnormal *Wolbachia* genomes, and then candidate abnormal *Wolbachia* genomes were further investigated by searching NCBI to filter out abnormal *Wolbachia* genomes. The mashtree v1.2.0 [64] was used to construct a large-scale NJ phylogenetic tree of the normal *Wolbachia* genomes (Appendix A). The ANI between *w*FI and other *Wolbachia* genomes was calculated using fastANI v1.33 [56], and the results were visualized using the Interactive Tree Of Life (ITOL) v6 [82].

We selected *w*FI and 25 complete *Wolbachia* genomes from the A–F supergroups (Appendix A). Multiple sequence alignments for single-copy genes were performed using Muscle v5.1 [79]. Poorly aligned regions were further trimmed using Trimal v1.4.1 [80]. We used ModelFinder [83] to obtain the best amino acid substitution model based on Bayesian Information Criteria. The phylogenetic tree was constructed using IQ-TREE v2.2.0.3 [81] with ultrafast bootstrap mode and 5000 iterations; *w*CfeT was used as an outgroup. Branch support was estimated using a Shimodaira–Hasegawa (SH)-like approximate likelihood ratio test with 1000 replicates. Finally, the maximum-likelihood phylogenetic tree was visualized using the ITOL v6 [82]. We also used the R package pheatmap v1.0.12 [84] to display the ANI between 26 complete *Wolbachia* genomes with a heat map.

## 5. Conclusions

We have successfully provided the complete assembly of the *Wolbachia* genome infecting *F. intonsa*, which belongs to supergroup B. Our genomic analyses have revealed several essential features, including mobile genetic elements such as prophage and ISs, as well as genes related to ANK, T4SS, riboflavin, and biotin synthesis and metabolism. The discovery of a complete riboflavin pathway in *w*FI suggested possibilities for meeting the riboflavin needs of the host. Additionally, the identification of an incomplete biotin synthesis pathway indicated potential interactions between *w*FI and the host or other endosymbionts to meet the host’s biotin needs. The first complete genome of the *Wolbachia* endosymbiont of *F. intonsa* provides a valuable resource for future investigations of *Wolbachia*–host interactions, comparative genomics of *Wolbachia*, and phylogenetic relationships between different supergroups of *Wolbachia*.

## Figures and Tables

**Figure 1 ijms-24-13245-f001:**
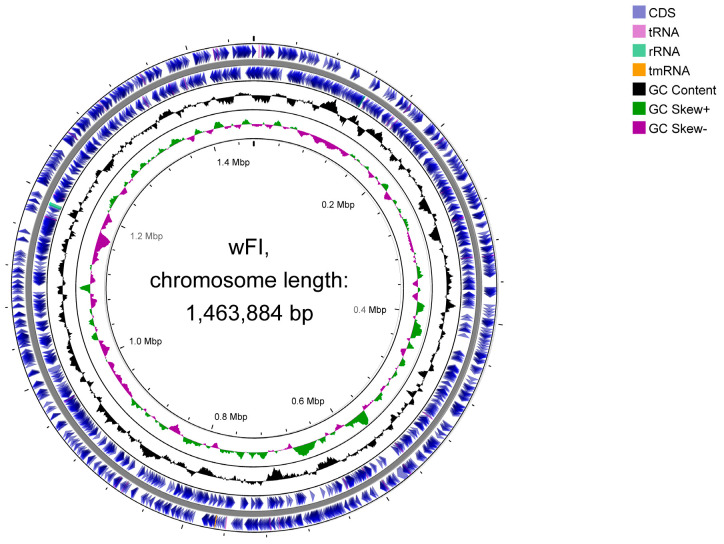
Circular map of the *w*FI genome. The outermost tracks 1 and 2 represent the positions of the CDS, tRNA, rRNA, and tmRNA genes on the positive and negative strands. Tracks 3 and 4 tracks show the GC content and GC skew, respectively.

**Figure 2 ijms-24-13245-f002:**
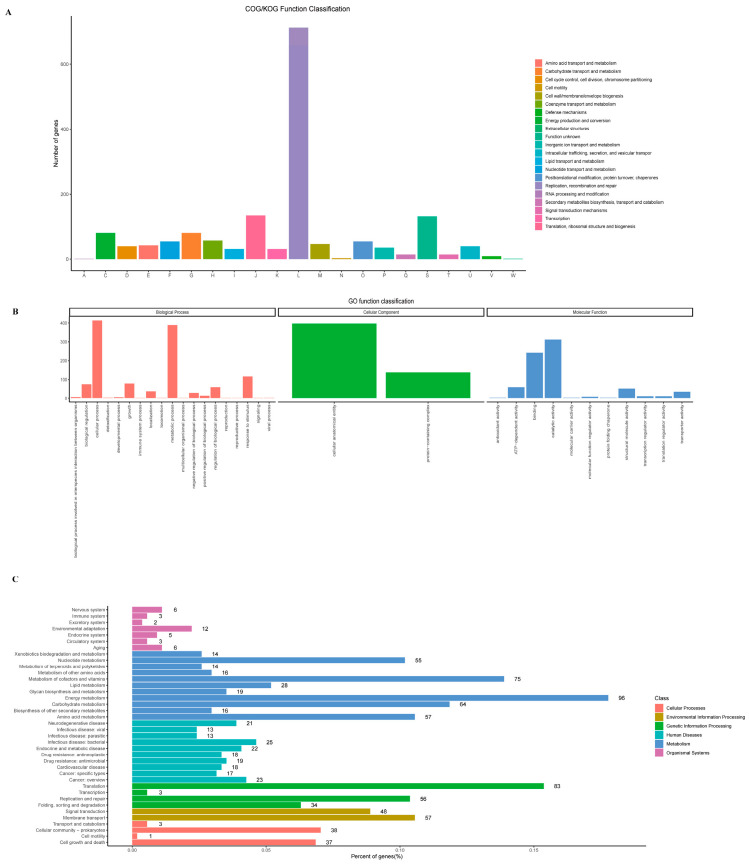
The functional genes annotation of the *w*FI genome. (**A**) COG functional annotation classification statistics. (**B**) GO function classification annotation. (**C**) KEGG pathway classification annotation.

**Figure 3 ijms-24-13245-f003:**
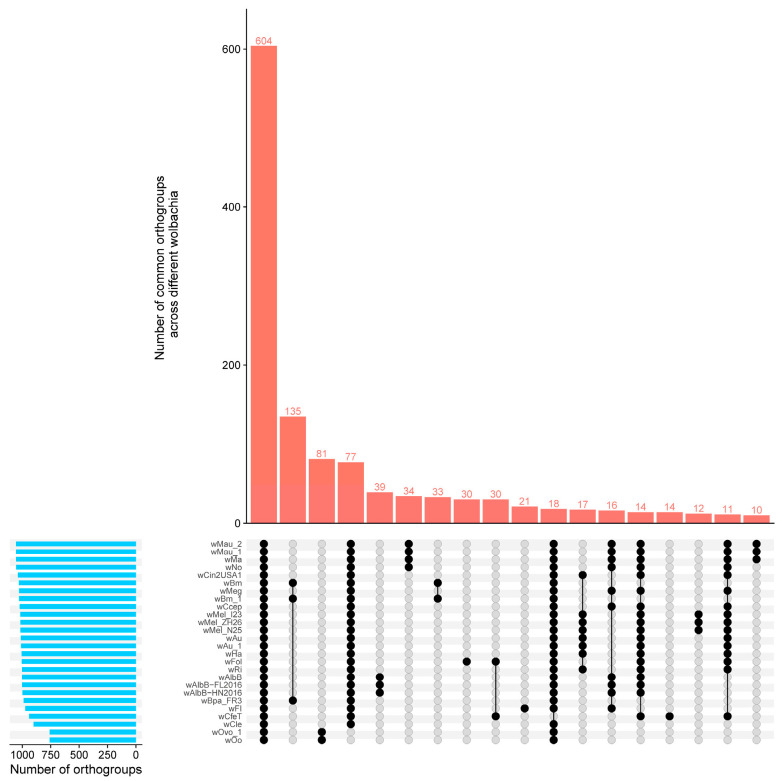
Analysis of protein orthology in complete genomes of *Wolbachia* belonging to supergroups A to F. The blue bar represents the number of orthogroups for the different *Wolbachia* strain. The red bar represents the number of common orthogroups across different *Wolbachia* strains, the first red bar represents 604 orthogroups across the 26 *Wolbachia* strains, the tenth red bar represents 21 *w*FI-specific orthogroups. Black dots indicate the presence, and gray dots indicate the absence of orthogroups in each *Wolbachia* strain.

**Figure 4 ijms-24-13245-f004:**
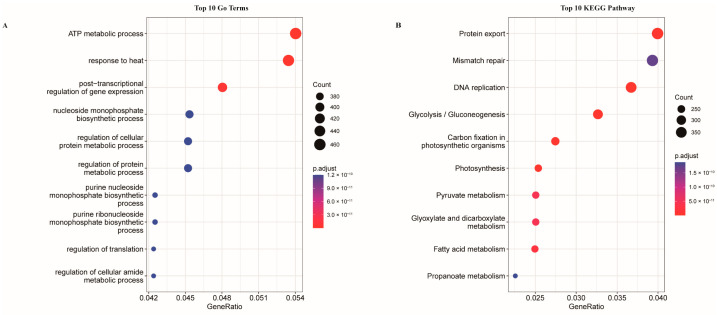
The GO and KEGG enrichment of the common orthogroups. (**A**) GO enrichment. (**B**) KEGG enrichment.

**Figure 5 ijms-24-13245-f005:**
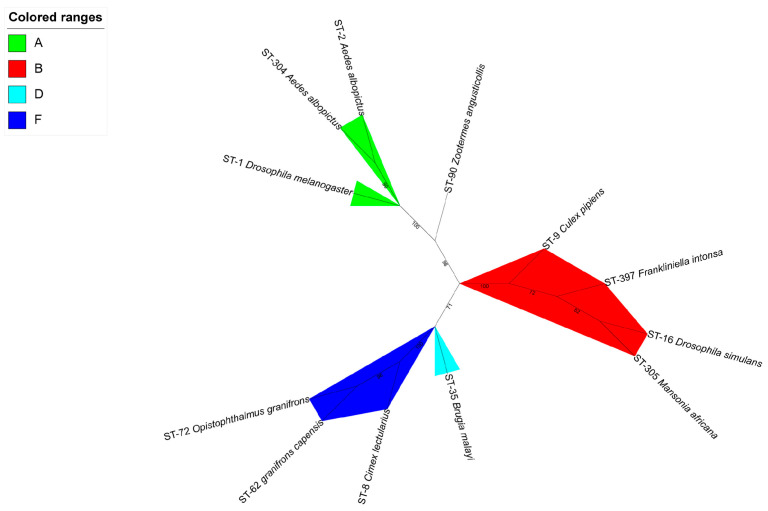
Maximum-likelihood phylogenetic tree of *Wolbachia* MLST from supergroups A, B, D, F, and H. The tree was generated using IQ-TREE v2.2.0.3 based on five MLST genes using ultrafast bootstrap mode with 5000 iterations. The amino acid substitution model used was GTR + F + G4. The *Wolbachia* supergroups are color-coded and shown in colored ranges.

**Figure 6 ijms-24-13245-f006:**
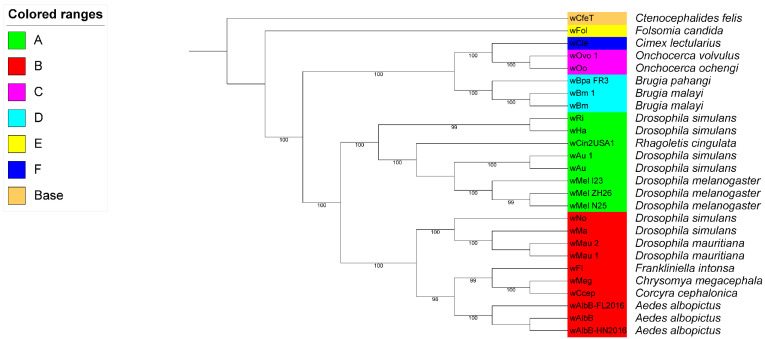
Maximum-likelihood phylogeny of the complete genomes of *Wolbachia* strains belonging to supergroups A to F, with *w*CfeT set as the outgroup. The tree was constructed using IQ-TREE v2.2.0.3 with ultrafast bootstrap mode, 5000 iterations. Branch support was estimated using a Shimodaira–Hasegawa (SH)-like approximate likelihood ratio test with 1000 replicates. The amino acid substitution model used was FLAVI + F + I + I + R4. Bootstrap values > 95% are shown at each node. The *Wolbachia* supergroups are color coded and shown in colored ranges.

**Figure 7 ijms-24-13245-f007:**
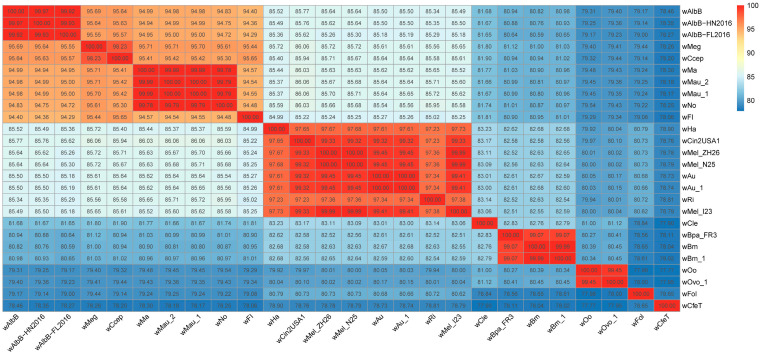
Heat map showing the ANI between 26 complete genomes of *Wolbachia* strains belonging to supergroups A to F. ANI were calculated using fastANI v1.33 with default parameters and visualized using the R package pheatmap v1.0.12.

**Table 1 ijms-24-13245-t001:** Key characteristics of *Wolbachia* genome in *F. intonsa* (*w*FI).

Attribute	*w*FI (GCA_029856955.1)
Size (bp)	1,463,884
Total genes	1877
Pseudogenes *	85
GC (%)	33.80
Proteins	1838
rRNAs	3
tRNAs	35
tmRNA	1
BUSCO Score	85.0%

* Candidate pseudogenes were mentioned in the log file rather than the result file.

## Data Availability

The data presented in the study are deposited in the GenBank database, accession numbers CP097148.

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
