# Peer review of "Complete De Novo Assembly of Wolbachia Endosymbiont of Frankliniella intonsa"

_ijms, 2023, doi:10.3390/ijms241713245_

Round 1

Reviewer 1 Report

The manuscript titled “Complete de novo assembly of Wolbachia endosymbiont of Frankliniella intonsa” is devoted to genome analysis of insects  endosymbiont, Wolbachia, that affects the host reproduction, immunity, and metabolism. This is the first genome reported for Wolbachia found Thysanopteran insects, such as Frankliniella intonsa. Wolbachia strain wFI has genome of 1,463,884 bp (GC content 33.80%) obtained by Nanopore long reads and Illumina short reads. The annotation of wFI identified a total of 1,838 protein-coding genes (includ-17 ing 85 pseudogenes), 3 ribosomal RNAs (rRNAs), 35 transfer RNAs (tRNAs), and 1 transfer-messenger RNA (tmRNA).

Beyond this basic description, Authors identified mobile genetic elements such as  prophage and insertion sequences (ISs), which make up 17% of the entire wFI genome, as well as 20 genes involved in riboflavin and biotin synthesis and metabolism. This research is important for understanding of nutritional mutualism between Wolbachia and the flower thrips, and for future studies delving into the intricate interactions between Wolbachia and its host.

It could be published in the present form, but lacks description of the host (Frankliniella intonsa) source and its preparation for sequencing  in the Materials and methods part.

Reviewer 2 Report

This manuscript describes a quality assembly of a Wolbachia species, which is an endosymbiont of Frankliniella intonsa. Thereafter there is a  brief analysis of the genome, endosymbiotic pathways and comparison with other genomes. The manuscript quality is high with only a few minor English typos:

The most abundant gene identified  ==> The most abundant Pfam profile identified 

large-sacle ==> large scale

the NCBI was searched 395 for filtering. ==> not clear what this means

ModelFinder [80] was used to ==> ModelFinder [80]  to 

iterations, using wCfeT was  ==> iterations, wCfeT was 

and its interac-417 tion with the host or other endosymbionts to meet the biotin requirements of the host. The  ==> unclear what this means

 Otherwise a very clear and straightforward manuscript to read.

This manuscript describes a quality assembly of a Wolbachia species, which is an endosymbiont of Frankliniella intonsa. Thereafter there is a  brief analysis of the genome, endosymbiotic pathways and comparison with other genomes. The manuscript quality is high with only a few minor English typos:

The most abundant gene identified  ==> The most abundant Pfam profile identified 

large-sacle ==> large scale

the NCBI was searched 395 for filtering. ==> not clear what this means

ModelFinder [80] was used to ==> ModelFinder [80]  to 

iterations, using wCfeT was  ==> iterations, wCfeT was 

and its interac-417 tion with the host or other endosymbionts to meet the biotin requirements of the host. The  ==> unclear what this means

 Otherwise a very clear and straightforward manuscript to read.
